# Single-Session Bilateral Genicular Artery Embolization for Knee Osteoarthritis via Brachial Access: A Case Report and Literature Review

**DOI:** 10.3390/diagnostics15172123

**Published:** 2025-08-22

**Authors:** Andrei Marian Feier, Florin Bloj, Octav Marius Russu, Andrei Bloj, Tudor Sorin Pop

**Affiliations:** 1Department of Orthopedics and Traumatology I, George Emil Palade University of Medicine, Pharmacy, Science, and Technology of Targu Mures, 540139 Targu Mures, Romania; andrei.feier@umfst.ro (A.M.F.); octav.russu@umfst.ro (O.M.R.); tudor.pop@umfst.ro (T.S.P.); 2Ares Excellence Center, Memorial Baneasa Hospital, 013812 Bucharest, Romania; bloj.florin@gmail.com; 3Department of Orthopaedics and Traumatology, Clinical County Hospital of Mureș, 540139 Targu Mures, Romania; 4Department 8, Radiology, Medical Imaging, and Interventional Radiology II, Carol Davila University of Medicine and Pharmacy, 030167 Bucharest, Romania

**Keywords:** knee osteoarthritis, genicular artery embolization, bioresorbable microspheres, angiogenesis, inflammation, bilateral intervention

## Abstract

**Background/Objectives:** Knee osteoarthritis (OA) significantly affects quality of life and poses substantial treatment challenges in patients with severe comorbidities that contraindicate total knee arthroplasty. Transarterial periarticular embolization (TAE) has developed as a minimally invasive alternative targeting pathological periarticular hypervascularity. Bilateral embolization in a single session has not yet been clearly documented. This case report describes the application of bilateral genicular artery embolization using bioresorbable gelatin microspheres. **Case report**: A 68-year-old male patient with severe bilateral knee OA and multiple cardiovascular comorbidities underwent simultaneous bilateral TAE using Nexsphere-F microspheres (100–300 µm). Embolization targeted hypervascular genicular branches identified through digital subtraction angiography preserving normal capsular and osseous perfusion. **Results**: At one-month follow-up, the patient’s pain score decreased dramatically (VAS from 8/10 to 2/10), accompanied by marked functional improvement (WOMAC score: from 64 to 84; KOOS score: from 49 to 72). No intraoperative or postoperative complications occurred and the patient required no analgesics post-procedure. **Conclusions**: Bilateral, same-session genicular artery embolization using bioresorbable gelatin microspheres provided short-term clinical benefits in a patient with advanced knee OA contraindicated for surgery.

## 1. Introduction

Osteoarthritis (OA) is the most common degenerative joint disease worldwide, impacting quality of life and functional independence [1]. Knee OA accounts for a substantial proportion of disability and economic burden among elderly populations [2]. The global incidence of knee OA is rising, paralleling aging populations, and increased obesity prevalence including lifestyle factors such as physical inactivity [3]. While total knee arthroplasty (TKA) is still the gold standard treatment for advanced knee OA, several patients present comorbidities and are ineligible for surgical intervention. Conditions such as cardiovascular disease, diabetes mellitus, severe obesity, and peripheral vascular disease elevate the risks associated with major orthopedic surgery and create a patient cohort in need of a viable non-surgical therapeutic alternative [4,5,6]. Interventional radiology has introduced transarterial periarticular embolization (TAE) as a minimally invasive alternative for managing symptomatic knee OA. TAE targets pathological periarticular hypervascularity which contributes significantly to chronic joint pain via sensory nerve proliferation and local inflammatory mediator release [7,8]. Selective embolization of hypervascularized periarticular tissues reduces inflammation and was proven to alleviate chronic pain and improve joint functionality without substantial periprocedural risks [9].

Previous investigations, including our initial experiences, demonstrated promising outcomes in knee OA patients [10]. We previously utilized imipenem/cilastatin sodium (IPM/CS) as an embolic agent with beneficial results regarding pain reduction and functional improvement [10]. The current literature highlights the importance of exploring alternative embolic agents to optimize clinical outcomes and cost-effectiveness [11]. Bioresorbable gelatin microspheres are an innovative embolic agent offering advantages: consistent particle size distribution, predictable vascular occlusion, almost immediate resorption and potentially lower rates of non-target embolization [12,13].

The present report details a unique case involving bilateral knee TAE performed in a single session using Nexsphere-F microspheres as an embolic agent. Bilateral procedure via brachial approach with bioresorbable gelatin microspheres represents a novel implementation not yet reported in the existing literature.

## 2. Case Report

This study was conducted in accordance with the Declaration of Helsinki and its subsequent amendments. Ethical approval was obtained from the Ethics Committee of Memorial Baneasa Hospital (approval code: 745300/07.08.2025). Written informed consent was obtained prior to the intervention.

### 2.1. Patient History and Assessment

The patient was a 68-year-old male with a body mass index of 31.3 kg/m^2^, employed as a warehouse manual handler. His medical history included type II diabetes mellitus, hypertension, NYHA class III congestive heart failure, bilateral carotid atherosclerotic plaques and partially occlusive stenosis of the proximal one-third of the superficial femoral artery. He denied any prior history of knee trauma. Clinical examination revealed a bilateral varus deformity. He presented with a progressive bilateral knee pain gradually increasing in the past six years. Clinical and radiological assessments confirmed end-stage OA. Radiographic evaluation showed bilateral Kellgren–Lawrence grade IV OA, affecting both medial and lateral compartments, with pronounced joint space narrowing, tibial marginal osteophyte formation, and subchondral sclerosis. The patient reported severe chronic pain (visual analogue scale [VAS] score: 8/10), recurrent monthly episodes of joint swelling, knee instability, and significantly impaired function, as evidenced by a WOMAC score of 64 and a KOOS score of 49. The patient’s symptoms had been refractory to multiple conservative therapies, including oral NSAIDs (etoricoxib and others), opioids, physical therapy, and intra-articular injections with corticosteroids and hyaluronic acid. TKA was contraindicated due to cardiovascular comorbidities. Due to severe, persistent symptoms and the inability to proceed with surgical intervention, TAE of the genicular arteries was proposed as a further therapeutic approach. The procedure was performed bilaterally in a single session.

### 2.2. Periarticular Embolization Procedure

Arterial access was obtained under local anesthesia via left brachial artery puncture, and a 5F radial introductory sheath was introduced. A 4 Fr multipurpose catheter (125 cm length) was navigated into the superficial femoral artery under fluoroscopic guidance. Initial digital subtraction angiography demonstrated pathological hypervascularity within the targeted periarticular regions of the knee joint, confirming active synovial neovascularization consistent with chronic inflammation (Figure 1a and Figure 2a).

A microcatheter (Direxion, 0.021-inch; Boston Scientific, Marlborough, MA, USA) was selectively advanced into each targeted genicular artery supplying the pathological periarticular vascular networks, the medial superior (Figure 1a) and lateral inferior (Figure 2a) genicular branches. To optimize procedural safety and vessel targeting, 200 mcg of intra-arterial nitroglycerine was administered to prevent vasospasm. Embolization was performed using a suspension of Nexsphere-F^TM^ bioresorbable gelatin microspheres (100–300 µm, NextBiomedical, Yeonsu-gu, Incheon, Republic of Korea), mixed with an iodinated contrast medium (Visipaque^®^; GE Healthcare Limited, Chalfont St. Giles, UK). Microspheres were selectively injected into the identified hypervascular branches under real-time fluoroscopic monitoring until achieving significant devascularization of the pathological synovial regions, as demonstrated by a notable reduction in angiographic hypervascular blush (Figure 1b and Figure 2b). Care was taken throughout the procedure to preserve normal vascular supply to adjacent capsular and osseous tissues, with no signs of unintended embolization. Completion angiography confirmed successful bilateral embolization, with optimal devascularization of hyperperfused synovial tissues and demonstrating preserved perfusion to adjacent healthy tissues (Figure 1b and Figure 2b). Hemostasis was achieved effectively through manual compression at the brachial access site. The patient tolerated the bilateral procedure without any intraoperative complications and was safely discharged four hours post-intervention with immediate ambulation and reduction in pain. Procedural metrics were as follows: total procedure time 12 min; total iodinated contrast 67 mL. Radiation exposure (conservative) was fluoroscopy time 21 min, cumulative air kerma 150–300 mGy, and dose–area 20 Gycm^2^, corresponding to an effective dose of 0.02–0.06 mSv.

### 2.3. Follow-Up and Results

Immediately after the procedure, the patient reported significant reduction in pain, with a VAS of 1 (slight discomfort while walking). However, the patient reported a reduction in knee pain with the VAS score decreasing from 8/10 pre-procedure to 2/10 at one-month follow-up. Functional scores improved with the WOMAC score increasing from 64 to 84, and the KOOS score improving from 49 to 72. No procedural complications or adverse events were reported during the follow-up period. The patient expressed general satisfaction with the outcomes, reporting improved mobility and no intake of analgesic medications in the first month.

## 3. Discussion and Literature Review

This case reports the efficacy and safety of bilateral TAE in managing symptomatic knee OA in a patient with significant cardiovascular comorbidities and contraindication to TKA, utilizing Nexsphere-F bioresorbable microspheres. The literature is scarce and existing evidence is not heterogenous.

### 3.1. Pathophysiological Basis for Targeting Hypervascularization via TAE

Chronic knee OA is marked by persistent joint inflammation and synovial proliferation, contributing significantly to perceived pain. Central to this process is angiogenesis driven by upregulated proinflammatory mediators (vascular endothelial growth factor) and TNF-α, stimulating aberrant neovascularization and sensory nerve growth [14,15]. Newly formed vessels increase chronic inflammation and synovitis with heightened nociceptive signaling within the periarticular tissue [3]. TAE selectively targets this pathological hypervascularization, disrupting blood supply locally and reducing inflammation-mediated pain signaling. This pathophysiological process supports the effectiveness of TAE as demonstrated by prior studies [7,16]. Clinical observations suggest there are two phases of pain relief after embolization: an immediate phase (within minutes to days) likely due to reduced blood flow and sensory nerve deactivation, and a delayed phase (weeks to months) as chronic synovitis abates following devascularization [17]. This biphasic response aligns with the proposed mechanism of breaking the inflammation–angiogenesis–pain cycle.

### 3.2. Comparison of Bioresorbable Gelatin Microspheres with Other Embolic Agents

Diverse embolic agents have been used in joint embolotherapy, including but not limited to IPM/CS, Embozene, and TANDEM microspheres [11]. Nexsphere-F resorbable microspheres used in our case combine uniform particle size and complete biodegradability, minimizing risks associated with permanent vascular occlusion [18]. The state of the art indicates promising short-term outcomes for these bioresorbable gelatin microspheres with balanced embolic effectiveness and safety without compromising long-term vascular integrity [19]. This advantage was evident in our patient’s immediate symptom relief and lack of early procedural complications. NexSphere-F distinguishes itself from traditional embolic agents by its transient vessel occlusion with rapid and predictable recanalization (Table 1).

In preclinical models, perfusion was restored as early as two hours post-embolization, with reduced inflammatory response and tissue infarction compared to permanent microspheres [20,21]. In contrast to permanent agents (tris-acryl gelatin microspheres or polyvinyl alcohol particles) which are associated with lasting occlusion and low recanalization but carry the risk of chronic ischemia and increased local inflammation [23,24], bioresorbable gelatin microspheres allow for controlled embolization with rapid biodegradation, preserving surrounding tissue viability on a long-term basis. One recent 155 patient study of Nexsphere-F reported a 67% mean pain reduction at 6 months post-embolization with no serious adverse events [30]. Only 6% of patients had post-embolization pain lasting beyond 1 week, a rate markedly lower than that observed with gelatin sponge particles in historical controls. Similarly, an alginate-based resorbable microsphere (SakuraBead) demonstrated > 75% improvement in WOMAC pain scores at 6 months in a pilot study proving that temporary embolics can yield substantial and sustained symptom relief in musculoskeletal conditions [31].

### 3.3. Evidence and Considerations for Bilateral Same-Session Embolization

Bilateral same-session genicular TAE is not yet documented in the literature but is justified by the frequent bilateral presentation of symptomatic knee OA in patients with severe comorbidities [32]. Previous genicular TAE series have included bilateral knee treatments [17] without reporting added complications. However, those cases were not described in detail as same-session procedures. Our report appears to be the first to deliberately perform and document single-session bilateral genicular TAE, adding to the evidence base for this method. This approach reduces cumulative procedural load, patient discomfort, and overall time to return to daily activities. Although concerns regarding extended procedural duration and the theoretical risk of contrast-induced nephropathy exist, these can be mitigated through meticulous technique and adequate periprocedural hydration [33]. Furthermore, Guo et al. demonstrated that renal function can be preserved in complex or prolonged embolization procedures when appropriate management strategies are employed, as shown by stable serum creatinine and eGFR values post-intervention [34]. Bilateral TAE also offers logistical advantages, reducing the need for repeat hospitalization and exposure to anesthesia. Our patient underwent a successful same-session bilateral embolization, which demonstrated clinical efficacy with no short-term adverse events. The use of brachial arterial access instead of a femoral approach allowed immediate post-procedural ambulation. Compared to transfemoral access, upper-extremity access routes (brachial or radial) reduce bleeding risk and enable faster mobilization [10]. Radial access is associated with the lowest complication rates [35], but vessel caliber and catheter length can limit its use for lower-extremity embolization [36]. Brachial access overcomes these technical constraints by accommodating larger catheters and providing robust support, with complication rates similar to femoral access when ultrasound guidance and careful hemostasis are applied [35]. Brachial access offered an optimal balance of technical feasibility, safety, and patient comfort in our same-session bilateral genicular TAE case.

However, prospective randomized controlled trials are necessary to evaluate whether bilateral genicular TAE can carry an increased risk of complications compared to the unilateral approach in a larger sample of patients.

### 3.4. Reported Clinical Outcomes

Prospective studies consistently reveal significant short-term improvements in VAS, WOMAC, and KOOS scores post-TAE [8,10]. Little et al. observed substantial WOMAC improvement (mean of 22 points) at one-month follow-up [8]. Okuno et al. demonstrated symptomatic relief with WOMAC improvements of approximately 10–15 points within one month after the procedure [14]. Our patient experienced similar functional and pain relief at one-month follow-up, reinforcing previous clinical observations of rapid onset symptom reduction after bilateral TAE. Short-term complications described in the literature include transient post-procedural pain, injection site discomfort, puncture site hemorrhage, transient cutaneous ischemia, and mild bruising [35]. Long-term concerns historically focus on theoretical risks such as avascular necrosis, chronic ischemic damage, decreased hip, or periarticular muscle strength and perineal paresthesia but are not scientifically proven yet [37]. The case we presented with bilateral procedure had no immediate or short-term complications, further supported by strict procedural protocols, including selective vasoconstriction (nitroglycerine, cold packs) with non-target vascular protection. Certain potential risks must be acknowledged in the context of genicular TAE and other musculoskeletal embolization procedures. Cumulative contrast load poses a concern in patients with pre-existing renal impairment or multiple comorbidities. While our case did not demonstrate renal deterioration, even prolonged procedures have been associated with stable renal function when hydration and reno-protective measures are employed. Current recommendations highlight minimizing contrast volume and tailoring it to the patients baseline renal function to mitigate the risk of contrast-induced nephropathy [34]. Moreover, vascular complications remain possible. Access-site hematomas are the most frequently reported, typically resolving spontaneously, whereas arterial spasm or minor microcatheter-related vessel injury may also occur but are usually transient and clinically insignificant. Rarely, larger hemorrhages or pseudoaneurysm formation have been described in the embolization literature, though not prominently in published GAE series [38]. Delayed ischemic events can present as transient cutaneous ischemia or mild soft tissue injury [34]. These effects are generally self-limited, resolving within days to weeks, though isolated cases of small cutaneous ulcers or asymptomatic bone infarcts have been observed in follow-up imaging. Importantly, no cases of clinically significant osteonecrosis, cartilage necrosis, or critical limb ischemia have been reported after GAE when performed selectively [16,31]. Long-term durability is still an evolving area of research. Our successful outcome suggests that bilateral genicular TAE is feasible in carefully selected patients. The clinical efficacy demonstrated here must be interpreted cautiously, as improvements may be temporary and durability beyond the short-term remains unproven. Prospective trials should compare bilateral vs. staged unilateral GAE to confirm that the convenience of a single-session approach does not come at the cost of increased complications or contrast load. Our case can serve as a proof of concept while we await larger studies. As this is a single case report with only one-month follow-up, the durability of symptom relief and the long-term safety profile cannot be established. The improvements observed in pain and function are temporary. These findings should therefore be interpreted with caution until supported by larger series with extended follow-up. Okuno et al.’s longitudinal data suggest sustained pain and functional improvements up to 24 months after unilateral TAE [9]. Comprehensive data beyond two years remain limited with the need for longer term evaluations. The current case provides immediate evidence of short-term benefits, but longitudinal follow-up is warranted. Strategic follow-up trials must include serial clinical assessments and repeat imaging to evaluate vascular viability or persistent ischemic signs. In particular, dedicated imaging follow-up with modalities such as MRI or Doppler ultrasound should be incorporated to assess synovial inflammation regression, detect subclinical ischemic changes, and confirm vascular safety after embolization. The absence of such imaging in the present case is a limitation, and we recommend its integration in future clinical protocols and research designs. Follow-up MRI studies have shown significant reduction in synovitis after bilateral TAE, correlating with pain improvement [17]. This suggests that embolization is not merely blunting pain signals but actively reversing inflammatory changes in the joint. Some authors have even hypothesized that by reducing synovial disease activity, bilateral genicular TAE could slow structural progression of OA or delay the need for arthroplasty [39]. Compared to radiofrequency ablation (RFA), intra-articular corticosteroid/hyaluronic acid injections, platelet-rich plasma (PRP), and subchondroplasty, TAE targets periarticular hypervascularity rather than direct joint structures or nerve conduction pathways [40]. RFA reduces nociceptive signaling temporarily and offers shorter symptom relief duration and repeated intervention necessity [41]. Intra-articular corticosteroids and PRP provide transient symptom control without addressing hypervascular inflammatory processes [42]. Subchondroplasty targets subchondral bone defects without addressing periarticular inflammation [43]. TAE demonstrates longevity in symptom relief and minimal reintervention rates with superiority in patients contraindicated for surgery or intolerant of repeated minimally invasive procedures. Comprehensive outcome comparisons remain necessary to delineate and establish optimal indications. TAE has potential to reduce healthcare costs compared to invasive surgical treatments. Reduced dependency on opioids and analgesics post-procedure can increase the cost-effectiveness by reducing opioid-related complications and chronic analgesic use risks [44]. Economic modeling studies indicate significant overall health system savings in patients with significant comorbidities who require prolonged preoperative optimization or postoperative rehabilitation if treated surgically [45].

### 3.5. Patient Selection Criteria and Alignment with the Literature

Previous TAE studies include patients unsuitable and/or unwilling to undergo surgery due to significant cardiovascular risk, surgery risk due to advanced age, obesity, and uncontrolled type II diabetes [10,38]. Our patient matched the literature-defined selection criteria: advanced OA, significant cardiovascular comorbidities contraindicating surgery, and obesity (BMI > 30), with prior unsuccessful conservative management. Further research should aim to focus on randomized controlled trials comparing Nexsphere microspheres versus conventional agents (IPM/CS, Embozene, TANDEM) to evaluate comparative outcomes comprehensively. Long-term prospective registries with periodic MRI follow-ups can assess durability and vascular recanalization along with associated late ischemic complications. Formal economic evaluations assessing resource utilization and costs are necessary to support TAE within clinical guidelines in this population group. Expanding patient selection criteria research to include early intervention in mild-to-moderate OA can uncover additional patient populations who could benefit from TAE.

## 4. Conclusions

Same-session bilateral genicular artery embolization with bioresorbable gelatin microspheres (Nexsphere-F) was performed in a patient with advanced knee osteoarthritis and severe cardiovascular contraindications to surgery. Our unique contribution lies in demonstrating the feasibility and short-term clinical benefit of a bilateral procedure performed via brachial approach in a single high-risk patient. Immediate improvements in pain and function suggest that this strategy can effectively manage symptoms and improve quality of life in carefully selected individuals. Bioresorbable embolic agents have been highlighted as advantageous compared to permanent agents, as they preserve long-term vascular integrity, minimize chronic ischemia risk, and provide a favorable balance between safety and efficacy. Nevertheless, these observations are preliminary and limited to a single case and should not be generalized to larger patient populations. Future work should include randomized controlled trials comparing bilateral with staged unilateral embolization, as well as prospective registries tracking long-term outcomes and imaging biomarkers, to establish the safety, durability, and broader applicability.

## Figures and Tables

**Figure 1 diagnostics-15-02123-f001:**
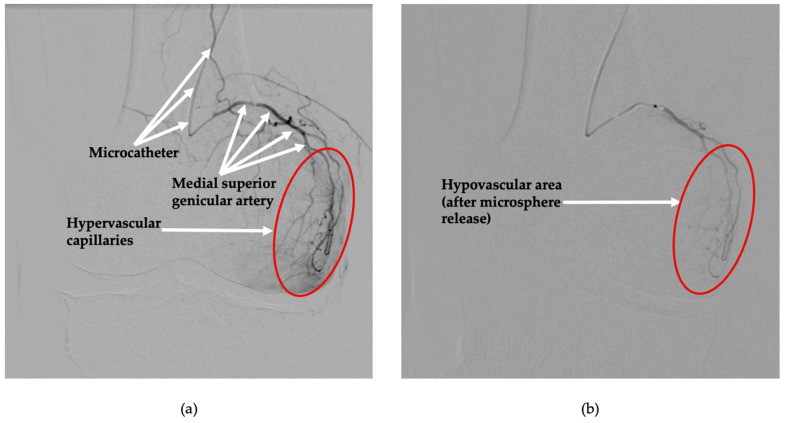
Digital subtraction angiography of the right knee before and after embolization of the medial superior genicular artery. (**a**) Baseline angiographic image with the microcatheter positioned within the medial superior genicular artery. Prominent hypervascular capillaries are observed within the anteromedial compartment of the knee (red circle), indicative of pathological synovial neovascularization. (**b**) Post-embolization image obtained following selective intra-arterial administration of microspheres. A visible reduction in vascularity (red circle) is seen within the previously hyperperfused region of targeted synovial tissue.

**Figure 2 diagnostics-15-02123-f002:**
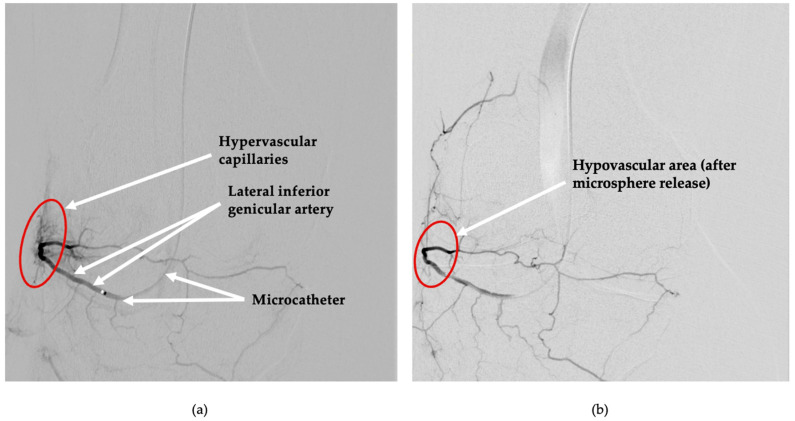
Digital subtraction angiography of the right knee before and after embolization. (**a**) Pre-embolization angiographic image demonstrating selective catheterization of the lateral inferior genicular artery via a microcatheter. A focal cluster of pathological neovascularization is visualized in the lateral periarticular region of the knee (red circle). (**b**) Post-embolization image showing a reduction in vascularity (red circle)within the previously hyperperfused zone following administration of resorbable microspheres indicating devascularization of the target region.

**Table 1 diagnostics-15-02123-t001:** Overview of embolic agent composition, occlusion behavior, and inflammatory response.

Agent	Resorbability	Vessel Occlusion and Precision	Inflammation and Recanalization
Resorbable gelatin microsphere (e.g., NexSphere-F^®^ R-GM)	- temporary [20]	- temporary occlusion - rapid recanalization (~2 h) [20,21]	- minimal tissue infarction (5.8% vs. 93% in control) - lower inflammatory cell infiltration and fibrosis in animal models [20,21]
Tris-acryl gelatin microspheres (TAGM e.g., Embosphere^®^)	- permanent occlusion - non-biodegradable [22]	- uniform calibrated size - lodges predictably in vessels of matching size [23]	- minimal inflammation - recanalization rare - persistent infarct areas [22,23]
Polyvinyl alcohol particles (PVA)	**-** permanent occlusion - non-biodegradable [24]	- irregular PVA—clumps and occludes proximally - spherical PVA—less predictable than TAGM [23,24]	- higher inflammatory response vs. TAGM - variable recanalization risk [23,24]
Gelatin sponge (e.g., Gelfoam, Marine Gel^®^)	**-** temporary (1–6 weeks) [25]	- particle size is variable - less distal penetration - less precise occlusion than TAGM [26,27]	- moderate inflammation - less extensive infarct area [28]
Liquid embolics (e.g., NBCA, Onyx)	**-** permanent [29]	- flows through complex vascular areas - ideal for arteriovenosus malformations - very precise occlusion - high risk of non-target spread [29]	- minimal recanalization - potential for non-target embolization - rare systemic toxicity [29]

## Data Availability

The data presented in this study are available on request from the corresponding author. Data are not publicly available due to privacy or ethical restrictions.

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
