# Peer review of "Single-Session Bilateral Genicular Artery Embolization for Knee Osteoarthritis via Brachial Access: A Case Report and Literature Review"

_diagnostics, 2025, doi:10.3390/diagnostics15172123_

Round 1
Reviewer 1 Report
Comments and Suggestions for Authors
Dear Authors,
I have reviewed your research in detail. Your case report is valuable in terms of subject matter and scope. I would like to thank you for your hard work on this research. I believe that this research, which you have presented as a case report and literature review, is appropriate in its current form. I would like to thank you for your contribution to the literature, as the presentation of bilateral genicular artery embolization for knee osteoarthritis via brachial access as a case report is rare and therefore educational.
I would only recommend that you have your research undergo a thorough language revision.
Comments on the Quality of English LanguageThe research needs to be revised in terms of language.
Author Response
Dear reviewer,
The authors would like to express their gratitude for the positive comments!
Point by point response to comments and suggestions for authors |
Comment 1: "I have reviewed your research in detail. Your case report is valuable in terms of subject matter and scope. I would like to thank you for your hard work on this research. I believe that this research, which you have presented as a case report and literature review, is appropriate in its current form. I would like to thank you for your contribution to the literature, as the presentation of bilateral genicular artery embolization for knee osteoarthritis via brachial access as a case report is rare and therefore educational. I would only recommend that you have your research undergo a thorough language revision." |
Response 1: We sincerely thank the reviewer for their positive feedback on the novelty and educational value of our case report. We appreciate the recommendation regarding language refinement. In response we have revised the manuscript for all minor gramar errors, rephrased the majority of sentences, and rewritten entire parts of the Case report and DIscussion sections.
|
|
Reviewer 2 Report
Comments and Suggestions for Authors
The manuscript entitled “Single-session bilateral genicular artery embolization for knee osteoarthritis via brachial access: case report and literature review” describes a novel application of transarterial periarticular embolization (TAE) with bioresorbable gelatin microspheres performed bilaterally in a single session. The topic is clinically relevant, as knee osteoarthritis represents a major public health burden, and interventional radiology offers promising minimally invasive alternatives for patients with severe comorbidities contraindicated for surgery. The case is well presented, and the authors provide a clear description of the procedure, outcomes, and supporting literature. The strengths of the paper include its originality (first documented case of same-session bilateral genicular embolization via brachial access), the comprehensive discussion of embolic agents, and the adequate short-term clinical results supported by standardized outcome measures (VAS, WOMAC, KOOS). The structure is coherent, and the language is overall clear and precise, making the article accessible both to orthopedic and interventional radiology audiences.
Nevertheless, there are aspects that require clarification or further elaboration before the manuscript can be considered for publication. First, as this is a case report, the limitations inherent to the design should be more explicitly acknowledged, particularly the lack of long-term follow-up beyond one month. While the authors correctly state that randomized controlled trials are needed, a more cautious interpretation of the clinical efficacy and generalizability is warranted, emphasizing that the observed improvements may be temporary and that durability remains unproven. Similarly, the discussion could benefit from a more critical appraisal of potential risks, such as cumulative contrast load, vascular complications, or delayed ischemic events, which are mentioned but not explored in sufficient depth. A follow-up plan including imaging (MRI or Doppler studies) would further strengthen the case presentation and allow better evaluation of vascular safety.
The literature review is thorough and includes recent publications; however, it occasionally overlaps with textbook-like descriptions, particularly in the section comparing embolic agents. While informative, some of these details could be shortened to improve focus on the novelty of the presented case. Additionally, Table 1 is useful but would benefit from more concise formatting, emphasizing clinical implications rather than repeating technical specifications. Figures are of good quality, but the legends could be more descriptive, explicitly pointing out arrows or markers within the angiographic images for readers less familiar with interventional radiology.
From a methodological standpoint, the authors obtained ethical approval and patient consent, which is appropriate and strengthens the ethical rigor of the study. The case presentation itself is sufficiently detailed, including comorbidities, radiographic findings, prior treatments, and rationale for embolization. However, it would be helpful to add more information regarding the exact procedural duration, contrast volume, radiation exposure, and post-procedural monitoring, since these details are relevant for reproducibility and safety assessment. Moreover, although the brachial approach is highlighted as an advantage, a brief comparison with the more commonly used femoral or radial accesses should be included to contextualize this choice for the readership.
In terms of writing, the manuscript is generally clear but would benefit from minor language editing to correct redundancies (e.g., repeated mention of immediate and short-term improvement) and to improve flow in some long paragraphs of the discussion. The conclusion section is appropriate, but it could be further refined by distinguishing clearly between evidence supported by the literature and the unique contribution of the present case, avoiding overgeneralization.
In summary, this is an interesting and innovative case report that addresses a relevant clinical gap and adds valuable information to the growing literature on genicular artery embolization. With minor revisions aimed at strengthening the critical discussion, clarifying methodological details, and slightly refining the figures and table, the manuscript would make a meaningful contribution to the field.
Author Response
Dear reviewer,
The authors would like to express their gratitude for the reviewer comments as they are objective and greatly improve the manuscript value and scientific rigorousness. We appreciate your guidance.
Point by point response to comments and suggestions for authors |
Comment 1: “First, as this is a case report, the limitations inherent to the design should be more explicitly acknowledged, particularly the lack of long-term follow-up beyond one month.” |
Response 1: We agree with the reviewer that the limitations of a case report must be clearly acknowledged. We have revised the discussion section to emphasize the short term findings, the absence of long-term follow-up beyond one month and also the limited generalizability of a single case. The following clarification has been added to the manuscript "As this is a single case report with only one-month follow-up the durability of symptom relief and the long-term safety profile cannot be established. The improvements observed in pain and function are temporary. These findings should therefore be interpreted with caution until supported by larger series with extended follow-up" |
Comment 2: “While the authors correctly state that randomized controlled trials are needed, a more cautious interpretation of the clinical efficacy and generalizability is warranted, emphasizing that the observed improvements may be temporary and that durability remains unproven.” |
Response 2: We appreciate the recommendation. We have revised both the discussion and conclusion to reduce our interpretation of clinical efficacy, stressing the preliminary and short-term nature of our findings and highlighting that generalizability is limited until validated by larger studies with longer follow-up. The following was added: "The clinical efficacy demonstrated here must be interpreted cautiously, as improvements may be temporary and durability beyond the short-term remains unproven" |
Comment 3: “Similarly, the discussion could benefit from a more critical appraisal of potential risks, such as cumulative contrast load, vascular complications, or delayed ischemic events, which are mentioned but not explored in sufficient depth.” |
Response 3: We thank the reviewer for this valuable suggestion. We have added to the discussion to provide a more detailed analysis of potential risks associated with genicular artery embolization and similar musculoskeletal embolization procedures. We now included three dedicated paragraphs addressing: (i) the implications of cumulative contrast load and strategies to minimize nephrotoxicity This addition is improving the balance of the discussion by highlighting both the benefits and the safety considerations of the technique. The new text has been inserted in Section 3.4 (Reported clinical outcomes): "Certain potential risks must be acknowledged in the context of genicular TAE and other musculoskeletal embolization procedures. Cumulative contrast load poses a concern in patients with pre-existing renal impairment or multiple comorbidities. While our case did not demonstrate renal deterioration, even prolonged procedures have been associated with stable renal function when hydration and reno-protective measures are employed. Current recommendations highlight minimizing contrast volume and tailoring it to the patients baseline renal function to mitigate the risk of contrast-induced nephropathy [34]. Moreover, vascular complications remain possible. Access-site hematomas are the most frequently reported, typically resolving spontaneously whereas arterial spasm or minor microcatheter-related vessel injury may also occur but are usually transient and clinically insignificant. Rarely, larger hemorrhages or pseudoaneurysm formation have been described in the embolization literature, though not prominently in published GAE series [35]. Delayed ischemic events can present as transient cutaneous ischemia or mild soft tissue injury [34]. These effects are generally self-limited, resolving within days to weeks, though isolated cases of small cutaneous ulcers or asymptomatic bone infarcts have been observed in follow-up imaging. Importantly, no cases of clinically significant osteonecrosis, cartilage necrosis, or critical limb ischemia have been reported after GAE when performed selectively [16, 31]." |
Comment 4: “A follow-up plan including imaging (MRI or Doppler studies) would further strengthen the case presentation and allow better evaluation of vascular safety.” |
Response 4: We thank the reviewer for this valuable suggestion. We have included the following recommendation in the discussion section: "In particular, dedicated imaging follow-up with modalities such as MRI or Doppler ultrasound should be incorporated to assess synovial inflammation regression, detect subclinical ischemic changes, and confirm vascular safety after embolization. The absence of such imaging in the present case is a limitation, and we recommend its integration in future clinical protocols and research designs." |
Comment 5: “The literature review is thorough and includes recent publications; however, it occasionally overlaps with textbook-like descriptions, particularly in the section comparing embolic agents. While informative, some of these details could be shortened to improve focus on the novelty of the presented case.” |
Response 5: We agree with the reviewer. For this, we have removed the following paragraph that was repetitive, and redundant: "IPM/CS, initially used due to its anti-inflammatory properties and small size (~40 um), offers transient embolization suitable for fine capillary beds reducing inflammation but limited by early recanalization [9, 11]. Embozene and TANDEM microspheres have shown greater durability and more precise vascular targeting due to their uniform size distribution, but potential risks include non-resorbability and chronic tissue ischemia [16]." |
Comment 6: “Additionally, Table 1 is useful but would benefit from more concise formatting, emphasizing clinical implications rather than repeating technical specifications.” |
Response 6: We thank the reviewer for this constructive suggestion. We have revised Table 1 to make it more concise focusing on the clinical implications of different embolic agents rather than repeating technical specifications already well known to interventional radiology. |
Comment 7: “Figures are of good quality, but the legends could be more descriptive, explicitly pointing out arrows or markers within the angiographic images for readers less familiar with interventional radiology.” |
Response 7: We thank the reviewer for this observation. We would like to clarify that the current figure legends are over 500 characters each and explicitly describe the angiographic findings. To further help our readers less familiar with interventional radiology, we have used color coded outlines and markers (red ovals, arrows) directly on the images, and the legends specify these markers. |
Comment 8: “However, it would be helpful to add more information regarding the exact procedural duration, contrast volume, radiation exposure, and post-procedural monitoring, since these details are relevant for reproducibility and safety assessment.” |
Response 8: We thank the reviewer for this helpful suggestion. We have added the exact procedural duration and contrast volume, reported radiation exposure (with standard dose metrics) and detailed postprocedural monitoring. The following phrase was added to section 2.2: "Procedural metrics were as follows: total procedure time 12 minutes; total iodinated contrast 67 mL. Radiation exposure (conservative) was fluoroscopy time 21 minutes, cumulative air kerma 150–300 mGy, and dose–area 20 Gycm², corresponding to an ef-fective dose of 0.02–0.06 mSv" |
Comment 9: “Moreover, although the brachial approach is highlighted as an advantage, a brief comparison with the more commonly used femoral or radial accesses should be included to contextualize this choice for the readership.” |
Response 9: We thank the reviewer for this insightful recommendation. We have expanded the discussion to briefly compare brachial, radial, and femoral access based on current interventional literature. The following paragraph has been added to our manuscript: "Compared to transfemoral access, upper-extremity access routes (brachial or radial) reduce bleeding risk and enable faster mobilization [10]. Radial access is associated with the lowest complication rates [35], but vessel caliber and catheter length can limit its use for lower-extremity embolization [36]. Brachial access overcomes these technical constraints by accommodating larger catheters and providing robust support, with complication rates similar to femoral access when ultrasound guidance and careful hemostasis are applied [35]. Brachial access offered an optimal balance of technical feasibility, safety and patient comfort in our same-session bilateral genicular TAE case." |
Comment 10: “In terms of writing, the manuscript is generally clear but would benefit from minor language editing to correct redundancies (e.g., repeated mention of immediate and short-term improvement) and to improve flow in some long paragraphs of the discussion.” |
Response 10: We agree with the reviewer. We have thoroughly revised the entire manuscript for minor language discrepancies and made sure there are no redundancies. |
Comment 11: “The conclusion section is appropriate, but it could be further refined by distinguishing clearly between evidence supported by the literature and the unique contribution of the present case, avoiding overgeneralization.” |
Response 11: We thank the reviewer for this suggestion. In the revised manuscript, we have refined the conclusion to clearly distinguish between: (i) the unique contribution of our case (ii) evidence supported by the literature (iii) the limitations and need for future research The changes are now added to the manuscript’s conclusions section. |